# VNIR-SWIR Spectroscopy, XRD and Traditional Analyses for Pedomorphogeological Assessment in a Tropical Toposequence

Jean J. Novais [1], Raúl R. Poppiel [1], Marilusa P. C. Lacerda [2] and José A. M. Demattê [1,*]

1 Department of Soil Science, Luiz de Queiroz College of Agriculture, University of São Paulo, Pádua Dias Av. 11, Piracicaba, P.O. Box 09, São Paulo 13416-900, Brazil; jeannovais@usp.br (J.J.N.); raulpoppiel@usp.br (R.R.P.)
2 Faculty of Agronomy and Veterinary Medicine, Darcy Ribeiro University Campus, University of Brasília, ICC Sul, Distrito Federal, P.O. Box 4508, Brasília 70910-960, Brazil; marilusa@unb.br
* Correspondence: jamdemat@usp.br; Tel.: +55-11-99767-0227

**Abstract:** Tropical climate conditions favor landscape evolution and the formation of highly weathered soils under different pedogenic processes due to certain differential properties. Traditional analysis coupled with VNIR-SWIR reflectance spectroscopy and X-ray diffractometry (XRD) analyses can reveal such characteristics. Several researchers cited throughout this study already discussed the possible applications of analyses in this field. All agree that integrated knowledge (holistic) can drive the future of the soil sciences. However, few refer to the potential of soil spectroscopy in deriving pedogenetic information. Thus, this paper aimed to assess pedomorphogeological relationships in a representative toposequence of the Brazilian Midwest using traditional analyses and geotechnologies. We performed landscape observations and soil sampling in the field. The laboratory's physical, chemical, spectral, and mineralogical determinations supported the soil classification according to the World Reference Basis (WRB/FAO) system. Based on the analysis results, we divided five profiles into two soil groups (highly and slightly weathered soils) using Pearson's correlation and hierarchical clustering analysis (HCA). Traditional analyses determined the diagnostic attributes. Spectroscopic readings from 0.35 to 2.5 µm wavelengths and XRD supported identifying soil attributes and properties. Finally, all soil classes were correlated according to correspondent reflectance spectra and primary pedological attributes. There was a strong correlation between spectral oxide features and X-ray diffraction peaks. The HCA based on oxide content and mineral composition validated the previous soil grouping. Thus, we could assess the pedomorphogeological relationships through VNIR-SWIR spectroscopy, XRD, and traditional analyses concerning pedogenic processes through their correlation with soil properties resulting from these processes. However, periodic measurements are required, making orbital sensing a continuous data source for soil monitoring.

**Keywords:** soil spectral behavior; pedogenesis; weathering; hydromorphic soils; hierarchical clustering analysis





## 1. Introduction

Soil formation factors provide distinctive characteristics that enable surveying soil properties, classification, and pedological mapping [1]. Soil attributes, such as color, texture, and mineralogy, for instance, are essential for studies of pedogenic processes [2–4], soil classification [5–8], and soil attribute mapping [9–11]. All this information is essential for agricultural engineering because it drives actions of management and localization for better productivity. Hence, reflectance spectroscopy, typically between the 0.35 and 2.5 µm spectral ranges, has enabled the inference of several chemical and physical attributes with satisfactory performance for soil science [12]. Traditional soil analyses require more resources and time, whereas spectroscopy is a cheaper, faster, non-destructive method that is

independent of chemical reagents. It can provide information on soil properties, simultaneously performing analyses with high productivity and accuracy [13]. Soil spectroscopy studies have improved soil science worldwide [14].

Soil attributes come from pedogenetic processes with a crucial role in agricultural practices, directly affecting the efficiency and sustainability of soil management [1]. A comprehensive understanding of these attributes is essential for informed decision-making and effective resource management [5]. For example: (a) Soil texture, i.e., the proportions of sand, silt, and clay particles, impacts water retention, permeability, and aeration, which are crucial for irrigation and root growth; Soil structure, i.e., the arrangement of particles into aggregates, influences water infiltration, root penetration, and overall porosity, affecting the void space within the soil and determining its water storage, drainage, and the oxygen availability to roots [9]; (b) Cation exchange capacity (CEC) indicates the soil's property to retain nutrients and make them available to plants, being fundamental for charge dynamics observation in soil solution [12].

Another key pedological attribute is the soil organic matter (SOM) [11]. It is the material formed by carbon-based remains of organisms in various degrees of decomposition, enhancing soil structure, nutrient retention, and biological activity. Additionally, soil minerals like clays and iron oxides affect chemical and physical properties, impacting fertility and nutrient release [12]. Soil sensing by spectroscopy across various wavelengths can determine these and many other soil attributes [13]. Therefore, studying such attributes enables tailored agricultural practices, optimizing resources, and ensuring long-term productivity and environmental health. In summary, studying soil attributes is central to agricultural engineering because it outlines the interactions between plant growth, water management, nutrient availability, and environmental equilibrium.

Based on these statements, studies have demonstrated the correlations between soil attributes at different wavelengths, which can aid in understanding pedogenic processes and support predictive models. Poppiel et al., 2019 [10] considered that the texture of tropical soils could be determined by spectroscopy and could be related to pedological classification. In this sense, we expect that the distribution, or absence, of different attributes and properties in soils are related to pedogenic processes, and that these can be identified through morphological interpretation of the reflectance spectrum [13].

Among its various applications, the Kubelka–Munk theoretical function is a technique for assessing spectra [14]. Processing of diffuse reflectance data can improve the prediction of soil constituents [15]. Soil reflectance data makes curves through the electromagnetic spectrum with absorption bands defined by minimum and maximum values around an axis at 0, by which specific features of minerals are easily identifiable [16].

Researchers agree that multivariate techniques simplify data structures by transforming one set of interdependent covariates into another, smaller set of independent covariates. Such multivariate statistics also classify samples, individuals, or covariates with similar characteristics [17]. For these authors, hierarchical clustering analysis supports soil evolution studies according to pedological attributes from a source material perspective, thus reinforcing this technique's validity for soil inferences.

As previously mentioned, we assume that traditional soil analyses supported by spectroscopy and X-ray diffractometry (XRD) can support discussions about the pedological evolution degree of long-diversified toposequences. Therefore, this paper aims to assess pedomorphogeological relationships by observing physicochemical attributes obtained through traditional analysis, XRD, and soil spectroscopy, considering pedogenic processes and soil management along a representative toposequence in the tropical environment.

## 2. Material and Methods

### 2.1. Study Area

The selected study area is a landscape representative of the Brazilian Central Plateau region, a remnant of the South American planing surface that developed during the tertiary geological period about 66 million years ago [18]. The toposequence had five pedological

profiles observed along 7200 m, starting from a high plateau where the weathering action decreases until the bottom of the hill. This area is in a watershed in the Distrito Federal, Brazil, from UTM coordinates 180,500 m to 190,845 m and 8,239,000 m to 8,230,600 m (Zone 23 South—Datum WGS-84) (Figure 1). Rocks of the Paranoá group, represented by the MNPpr3 slate unit (MNPpa and metarhythmite unit), form the local lithology [19]. They comprise two distinct geomorphological surfaces: 1. The High Plateaus Region (GS-1); 2. The Intermediate Dissection Area (GS-2) and Dissected Valley Region (GS-3), presenting a slope gradient ranging from flat (0–3%) to gently wavy (3–8%) [20].

Alvares et al., 2013 [21], when mapping the Brazilian climate, identified that the DF region corresponds to the CWA domain (high altitude tropical climate) through Köppen's climate classification [22], with a rainy season between October and April with precipitation varying from 1500 to 2000 mm (84% of the annual total). The local vegetation is classified as *Cerrado* (Savannah), characterized by low trees, sparse shrubs, and grasses, which are subdivided into several subtypes [23].

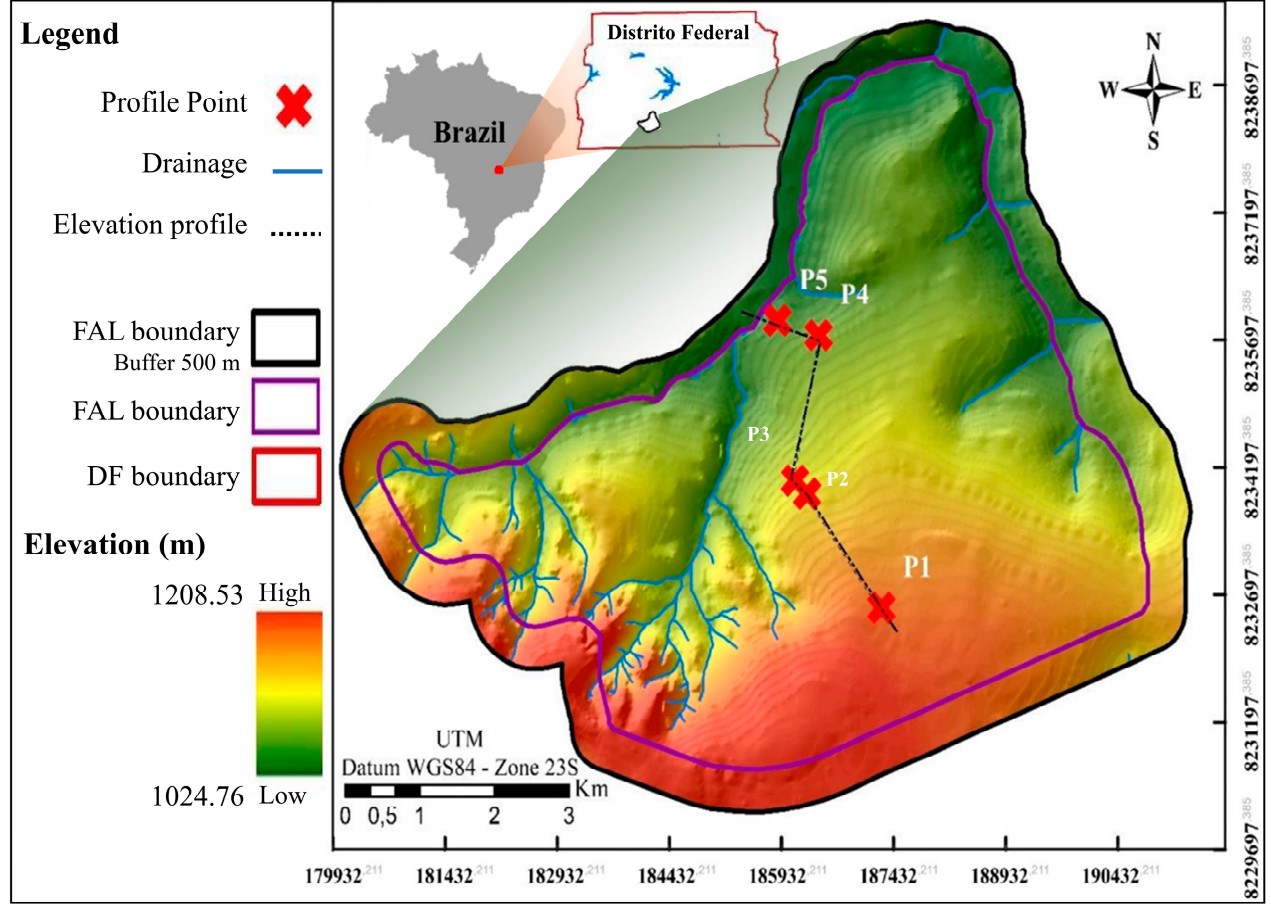

**Figure 1.** Localization map of the study area with elevation and sampling sites. Source: Adapted from Carvalho, Jr. et al., 2022 [24].

### 2.2. Morphological, Chemical, and Physical Analyses

We classified the soil through field and laboratory analyses according to the World Reference System [25]. Firstly, we performed morphological analyses in the field in October 2021, during which we analyzed soil attributes and properties such as color (using the Munsell Color Chart [26]), structure, and particle size [27].

Physical evaluations followed [27] for soil samples taken at the surface and subsurface horizons. Deformed soil samples were air-dried, crumbled, and sieved through 2 mm mesh for chemical and physical determination in the laboratory according to [28]. The

$Ca^{2+}$, $Mg^{2+}$, and $Al^{3+}$ levels were determined by potassium chloride (KCl) extraction. The exchangeable acidity ($H^+ + Al^{3+}$) was determined by a calcium acetate extractor. A Mehlich-1 extractor determined the $P_2O_5$ (phosphorus) contents while readings on a flame photometer determined K (potassium) contents. Wet oxidation determined the soil organic carbon content (SOM).

Undisturbed soil samples were taken in triplicate from the surface and subsurface horizons using a volumetric cylinder (100 $cm^3$) for the determination of gravimetric soil moisture (Ms) and density (Ds) [28]. Soil texture, clay, silt, and sand were determined using the Bouyoucos method with NaOH dispersion [29]. We also calculated the degree of flocculation (Fd) from the amounts of total clay (TC) and the particle density (Pd) using the volumetric flask technique [28,29].

### 2.3. X-ray Diffraction and Soil Reflectance Spectroscopy Data

X-ray diffractometry (XRD) analysis followed the recommendations of USDA, 2014 [28,30], which establishes the use of 50 g of air-dried and sieved soil samples at <2 mm (particles smaller than the fine sand fraction) to identify their mineral composition. After, we submitted soil samples to four treatments: (a) Total—20 g of air-dried fine soil was macerated; (b) Clay—20 g samples of macerated soil were placed in aqueous suspension for 8 h. After sedimentation, the soil supernatant portion was separated and subjected to sedimentation for 8 h. This procedure was repeated several times until it reached the required clay amount [31]; (c) Cold Glycolate—clay samples were treated with glycerine and ethylene glycol suspension and centrifugation; (d) Heating—part of the clay was sedimented by centrifugation. After this step, the samples were placed on microscope slides in an oven at 110 °C for 2 h. Finally, all the slides were analyzed by 3 to 40° XRD and interpreted by MDI's JADE software (Materials Data, Inc., Livermore, CA, USA) [32]. Measurements were obtained from the peak intensities of minerals in the Total and Clay treatments.

Thereafter, the air-dried soil fraction was placed on Petri dishes for reflectance spectroscopy analysis between the 0.35 and 2.5 µm spectral ranges using the FieldSpec® 4 Pro spectroradiometer (Analytical Spectral Device Inc., Boulder, CO, USA) [33], which is standardized to a 3 nm spectral resolution from 350 to 700 nm and 10 nm between 700 and 2500 nm, interpolated to 1 nm [13]. The Kubelka–Munk (K–M) theory establishes a correlation between absorption coefficients and dispersion with reflectance [14], as shown in Equation (1).

$$\frac{k}{s} = \frac{(1 - R_\infty)^2}{2R_\infty} \tag{1}$$

Here, *s* is the scattering coefficient, *k* is the absorption coefficient of the sample, and $R_\infty$ is the diffuse reflectance [14]. We applied K–M to obtain patterns in specific peaks of the diffuse reflectance in VNIR and SWIR intervals, which were amplified for Pearson's correlation between soil attributes and reflectance factors along wavelengths.

The results of the chemical, physical, mineralogical, and spectral analyses of 10 diagnostic horizons for each pedological class of the studied region were used to perform Pearson's correlation, spectral median, and hierarchical clustering analyses (HCA) of the local database of soil attributes (oxide content) [6,7]. This information was extracted from the Brazilian Spectral Soil Library—https://bibliotecaespectral.wixsite.com/english (accessed in 13th June 2023) database of the GeoCiS Research Group from Esalq/USP—https://esalqgeocis.wixsite.com/english (accessed in 23th June 2023).

We submitted the soil diagnostic horizon attributes and reflectance factors to HCA from the Euclidian distance similarity index on oxide content obtained from the region's database and spectral curve soil classes. We hypothesize that spectral data can explain pedogenesis by analyzing the interdependence between soil attributes and spectral behavior under the K–M function. The HCA by Ward's algorithm highlights groups with less variance between them [17].

### 3. Results

Five soil profiles, determined according to the FAO's global soil classification system [25], located in a toposequence over a distance of 7200 m and an altitude ranging from 1175 to 1050 m (Figure 2) showed differences in their properties, such as hydromorphism, which led us to divide them into two groups: Group I—highly weathered soils: P1—clayic, dystric, Rhodic Ferralsol; P2—clayic, dystric, Rhodic Ferralsol; P3—clayic, petroplinthic, dystric, Haplic Ferralsol. Group II—less weathered soils: P4—clayic, dystric, Haplic Gleysol; P5—dystric, hemic, Haplic Histosol. Notably, the different depths occurred due to the particularities of each profile, which had some natural barriers in the bottom of the trench, be it different layers, such as the petroplinthite within the profile; water, as found in P4; or the weathered rock in P5. The first two profiles did not present such impediments or differences up to 150 cm, being the deepest soils of the toposequence.

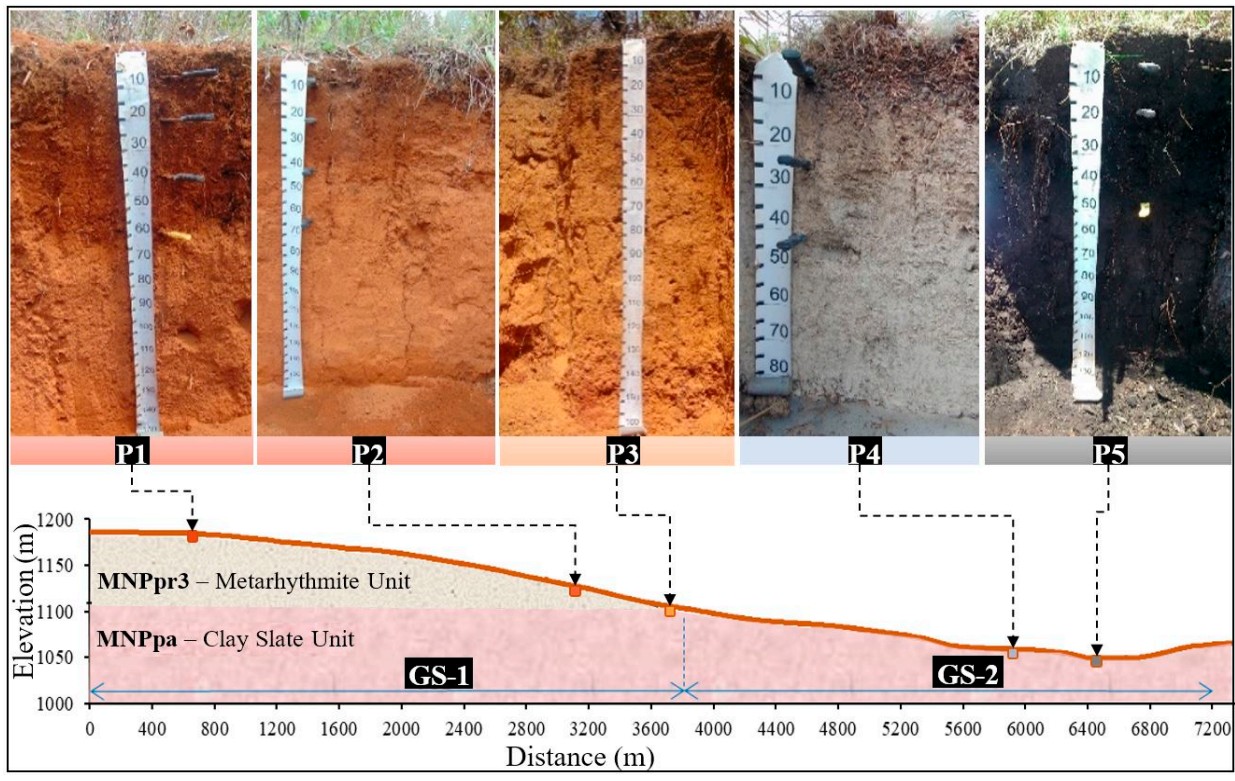

**Figure 2.** Topographic profile of the toposequence and the relative position of soil profiles. Source: Adapted from Carvalho, Jr. et al., 2022 [24]. (P) soil profile; (GS) geomorphological surface.

In terms of pedomorphogeological relationships, the first group consisted of P1, P2, and P3 subgroups; P1 was located at the upper level of the toposequence (~1175 m) in flat relief, followed by P2 at about 1120 m altitude with a gently wavy relief, next to the edge of a high plateau. Both profiles are in the GS-1 and consist of pelitic rocks, facies, pelitic intercalations of metarrythmits of slate, and metarrythmits of the Paranoá Group that are rich in iron (Fe) [19]. This way, the region's climatic conditions have promoted the landscape's evolution, favoring pedogenesis with different soil formation processes and diagnostic attributes. We have labeled all the results of the morphological, physical, and chemical analyses in the tables below (Tables 1 and 2). The discussion considers these values to support the argumentation. The pedological horizon of the profiles starts after a thin layer of non-decomposed organic material.

**Table 1.** Morphological and physical attributes of the FAL toposequence.

| Hz | Depth (cm) | Munsell Color Chart, 1992 [26] | Struct. | Clay | Silt | Sand | Text. | S/C | Ms | Ds | Dp | TP | Fd |
|---|---|---|---|---|---|---|---|---|---|---|---|---|---|
| | | | | | g kg⁻¹ | | | | g g⁻¹ | g cm⁻³ | | % | g g⁻¹ |
| | | | | | | | | | | | | | |
| A | 2–52 | 10R 3/6 | vs, g | 707.43 | 230.81 | 61.76 | vc | 0.32 | 24.69 | 0.72 | 2.35 | 68.13 | 91.89 |
| B | 52–160 | 10R 5/8 | vs, g | 726.69 | 215.43 | 57.87 | vc | 0.29 | 7.16 | 0.83 | 2.50 | 61.35 | 86.42 |
| | | | | | | | | | | | | | |
| A | 2–45 | 5YR 4/6 | vs, g | 678.65 | 229.02 | 92.33 | vc | 0.33 | 20.58 | 0.61 | 2.20 | 70.75 | 90.06 |
| B | 45–150+ | 2.5YR 4/6 | vs, g | 703.74 | 225.29 | 70.97 | vc | 0.32 | 9.45 | 0.79 | 2.60 | 68.56 | 84.25 |
| | | | | | | | | | | | | | |
| A | 3–48 | 7.5YR 4/4 | vs, g | 566.61 | 289.28 | 144.10 | c | 0.51 | 29.19 | 0.71 | 2.50 | 71.79 | 88.70 |
| B | 48–125 | 7.5YR 5/8 | vs, g | 611.38 | 281.77 | 106.85 | vc | 0.46 | 20.80 | 0.85 | 2.47 | 65.58 | 78.15 |
| B | 125+ | 7.5YR 4/8 var. 10R 3/6 | | | | | Concretions | | | | | | |
| | | | | | | | | | | | | | |
| A | 5–25 | 6/10Y | s, b | 473.70 | 312.31 | 213.99 | c | 0.66 | 25.89 | 0.72 | 2.33 | 59.90 | 89.73 |
| C | 42+ | 10YR 7/2 var. 5YR 6/8 | m | 655.09 | 249.47 | 95.44 | c | 0.38 | 30.78 | 0.91 | 2.41 | 50.98 | 92.32 |
| | | | | | | | | | | | | | |
| O1 | 3–18 | 10YR 2/2 | l, p | 218.61 | 240.03 | 541.36 | lsc | 1.10 | 56.66 | 0.26 | 1.72 | 84.03 | 92.01 |
| O2 | 18–50 | 10YR 2/1 | l, p | 210.70 | 256.29 | 533.01 | lsc | 1.22 | 65.91 | 0.45 | 1.96 | 78.99 | 100.00 |
| A | 50+ | 10YR 2/2 | l, p | 285.30 | 439.53 | 275.25 | lc | 1.54 | 59.71 | 0.72 | 2.22 | 54.63 | 93.13 |

P1—very clayey, dystric, Rhodic Ferralsol
P2—very clayey, dystric, Rhodic Ferralsol
P3—very clayey, petroplinthic, dystric, Haplic Ferralsol
P4—very clayey, dystric, Haplic Gleysol
P5—very clayey, hemic, Haplic Histosol

(Hz) horizon of the profile; (var.) variegated color; (Struct.) structure; (vs) very small; (s.) small; (l) large; (g) granular; (b.) blocky; (m) macice; (p) prismatic; (Text.) soil texture: (vc) very clayey; (c) clayey; (lsc) loamy–sandy–clayey (lc) loamy–clayey; (S/C) silt–clay ratio; (Ms) soil gravimetric moisture; (Ds) soil density; (Dp) particle density; (TP) total porosity; (Fd) degree of flocculation; (P) profile.

**Table 2.** Chemical attributes of the FAL toposequence.

| Hz | pH H₂O | Ca²⁺Mg²⁺ | Al³⁺ | H+Al³⁺ | K | P | SB | t | CEC | m | V | SOM |
|---|---|---|---|---|---|---|---|---|---|---|---|---|
| | | cmolc dm⁻³ | | | | mg dm⁻³ | | cmolc dm⁻³ | | | % | g kg⁻¹ |
| | | | | | | | | | | | | |
| A | 5.16 | 0.35 | 0.14 | 4.29 | 0.08 | 0.46 | 0.43 | 0.57 | 4.72 | 24.56 | 9.11 | 30.35 |
| B | 4.86 | 0.48 | 0.11 | 3.10 | 0.05 | 1.02 | 0.53 | 0.64 | 3.63 | 17.19 | 14.60 | 20.39 |
| | | | | | | | | | | | | |
| A | 4.57 | 0.74 | 0.22 | 6.63 | 0.06 | 0.63 | 0.80 | 1.02 | 7.43 | 21.57 | 10.77 | 40.51 |
| B | 4.88 | 0.60 | 0.12 | 3.98 | 0.06 | 0.77 | 0.66 | 0.78 | 4.64 | 15.38 | 14.22 | 20.14 |
| | | | | | | | | | | | | |
| A | 5.12 | 0.79 | 0.22 | 6.67 | 0.08 | 1.03 | 0.87 | 1.09 | 7.54 | 20.18 | 11.54 | 53.38 |
| B | 5.19 | 0.7 | 0.16 | 2.97 | 0.01 | 0.39 | 0.71 | 0.87 | 3.68 | 18.39 | 19.29 | 47.99 |
| Bc | 5.90 | 0.61 | 0.04 | 1.11 | 0.04 | 0.40 | 0.65 | 0.69 | 1.76 | 5.80 | 36.93 | 19.79 |
| | | | | | | | | | | | | |
| A | 4.68 | 1.08 | 0.63 | 8.51 | 0.17 | 1.12 | 1.25 | 1.88 | 9.76 | 33.51 | 12.81 | 78.10 |
| Cg | 5.48 | 1.38 | 0.45 | 2.22 | 0.18 | 0.85 | 1.56 | 2.01 | 3.78 | 22.39 | 41.27 | 22.45 |
| | | | | | | | | | | | | |
| O1 | 4.93 | 0.64 | 1.51 | 10.83 | 0.18 | 1.89 | 0.82 | 2.33 | 11.65 | 64.81 | 7.04 | 160.03 |
| O2 | 4.95 | 0.95 | 1.24 | 11.79 | 0.15 | 2.64 | 1.10 | 2.34 | 12.89 | 52.99 | 8.53 | 165.99 |
| A | 5.06 | 2.21 | 1.91 | 11.02 | 0.21 | 2.72 | 2.42 | 4.33 | 13.44 | 44.11 | 8.01 | 191.18 |

P1—very clayey, dystric, Rhodic Ferralsol
P2—very clayey, dystric, Rhodic Ferralsol
P3—very clayey, petroplinthic, dystric, Haplic Ferralsol
P4—very clayey, dystric, Haplic Gleysol
P5—very clayey, hemic, Haplic Histosol

(Hz) horizon of the profile; (SB) sum of exchangeable bases; (t) effective cation exchange capacity; (CEC) cation exchange capacity at pH 7.0; (m) saturation by aluminum; (V) base saturation; (SOM) soil organic matter in the soil; (P) profile.

## 4. Discussion

### 4.1. Morphological Attributes

Looking at soil attributes in the previous Results section, we observed that both Rhodic Ferralsols have a typically very small granular structure and reddish colors, ranging from 5YR to 10R hues, with values from 3 to 5 and chroma from 6 to 8, respectively, for P1 and P2, increasing in deeper layers. The local phytophysiognomy in P1 is the *Cerrado strict sense*, which is similar to a savanna formation [23], characterized by low trees and vegetation that

is slightly dense in its herbaceous layers; while P2 is categorized under sparse *Cerrado*, it is shrubbier than P1.

Haplic Ferralsol found in P3 has a B horizon that is less thick than P1 and P2 due to a concretionary horizon from a depth of 125 cm. Such a profile has been assigned at 1112 m elevation, adjacent to P2, towards the dissected edges in GS-2, under a gently wavy landscape on the same lithologies as the other Ferralsols. The variation in the color of the Group 1 soils is due to the parental rock, whose internal drainage flow is rapid enough to expose the iron present in the soil solution to oxidation, producing more reddish tones.

The red–yellowish color (7.5YR) of Haplic Ferralsol could be explained by the lowest value of hematite/goethite ratio (Ht/Gt) compared to other Ferralsols [6]. The obstructive layer formed by the concretionary horizon hinders internal drainage, keeping soil moisture in the profile longer and maintaining Gt, which is more stable than Ht. Therefore, petro-plinthite from the concretionary horizon restricts water infiltration and promotes changes in pedoclimate, soil compounds, and spectral behavior [4,13].

Therefore, the parental rock is not decisive for the color variation in P3, as in P1 and P2, in which Fe was more oxidized in the soil (e.g., higher Ht and Gt contents), coloring it [11]. On this occasion, P3 was under agricultural use. However, we could observe remnants of vegetation from the *Dirty Field–Cerrado* formation [23].

Moving on to the Group 2 soils, P4 had an elevation of 1071 m, was recorded in GS-2, and had a gently wavy relief at the base of the toposequence. The $C_{glei}$ horizon is a layer formed by the gleization pedogenic process [3]. This horizon showed dark greyish colors, 2.5/N and 6/10/N. As reported by Novais et al., 2021 [6], the gleization pedogenic process promotes the formation of small blocks in the $C_{glei}$ and A horizons. Besides the vegetation of flooded areas, the presence of *Trembleya parviflora*, a shrubby weed, was observed, indicating an environmental imbalance due to lowering groundwater.

Under reducing conditions, iron changes from $Fe^{3+}$ to $Fe^{2+}$, which is soluble, and it leaches out of the soil, giving it a blue–greyish color, as Fe oxides are responsible for reddish soil colors [6]. Terra et al., 2018 [3] stated that soils formed in a reductiomorphic environment have a greyish, greenish, or bluish appearance due to the absence of oxidized iron, which undergoes a gleization process. Novais et al., 2023 [7], who studied soils in DF, attributed the causes of soil colors to the predominance of hydromorphic conditions in the area. In addition, the transitional plinthic diagnostic horizon (R) showed variegated colors due to an oxidized and reduced Fe mixture.

The fifth profile, P5, was exposed in GS-2 under a flat relief at the bottom of the toposequence, with an altitude of 1050 m in the flooding area. The high SOM content causes strong darkish colors, with a hue of 10YR, and a value and chroma varying between 2 and 1. Higher SOM content causes large and prismatic structures in this soil's O and A horizons [6,7]. Regarding land use, the area was under pasture cultivation, with the remaining riparian forest preserved along the creek. Greater flood flow favors hydromorphism in the soil profile, which promotes SOM accumulation in the decomposition stages and confers high natural fertility to this soil [3]. P5 stays in a place of continual deposition of sediments and organic matter, forming a mineral–organic soil with a thicker, darkish, hemic-character layer, as reported by [25].

### 4.2. Physical Attributes

The physical attributes of the soil classes studied can also be seen in Table 1. It shows that the texture varies from very clayey in P1 and P2, changes to clayey in P3, and changes to loamy and sandy–loamy in P4 and P5, respectively. The density values (Ds) varied between 0.26 g cm$^{-3}$ for surface horizons of P5 and 0.91 g cm$^{-3}$ at subsurface horizons of P4 due to higher clay content, which caused discrete compaction in topsoil.

According to Poppiel et al., 2019 [10], Ferralsols from tropical regions have textures ranging from clayey to very clayey, while their Ds are around 0.7 and 1.0 due to high exposure weathering. Novais et al., 2021 [6] obtained similar results when analyzing soil in a region near the study area. The textures of the last two profiles are derived from the

input of sediments with different granulometry, featuring the addition pedogenic process. Therefore, the mineralogical composition may not be from the local parent rock.

The silt/clay ratio (S/C) for Group 1 indicated a higher degree of pedological evolution from P1 to P3. According to the Soil Survey Staff [28], soil horizons with S/C ratio values below 0.6 with clayey textures represent highly evolved soils. Consequently, S/C had a regular variation from 0.29 in P1 to 0.46 in P3. It is worth mentioning that this statement does not apply to Group 2 profiles (P4 and P5) because their texture is due to the accumulation of illuvial sediments and they are not necessarily related to parent rock.

Gravimetric soil moisture (SM) in P1 and P2 varied from 20.58 to 24.69 g g$^{-1}$ in surface horizons and from 7.16 to 9.45 g g$^{-1}$ in diagnostic subsurface horizons. With similar values, SM reached 29.19 on the surface and 20.80 g g$^{-1}$ on the diagnostic subsurface horizons for P3. SM in Ferralsol surfaces is also due to a high evolution degree, where Gt and Gb presence and the face-to-face adjustment of Kt plates, facilitated by a very small granular structure, retain some moisture [6]. For P4, the Ms values ranged from 25.89 to 30.48 g g$^{-1}$. The highest values found in P5 reached 59.71 g g$^{-1}$ on the A horizon, which is related to the high SOM associated with the deficit in the local drainage.

Pedoenvironmental conditions influenced high Ms values in the P3 subsurface horizon, which was attributed to the reduction in soil infiltration capacity by the concretionary horizon. It is worth mentioning that sampling occurred during the rainy season in DF, which explains high soil moisture values from specific conditions of water percolation into every profile. Such characteristics increase porosity (TP) and decrease Ds [28].

The Ds reveal the evolution degree and pedogenic processes because they result from solid particle arrangement in the soil. Therefore, D depends on the texture and SOM content [28,30]. Ds showed values ranging from 0.71 to 0.85 g cm$^{-3}$ in subsurface horizons and from 0.72 to 0.91 g cm$^{-3}$ in diagnostic horizons. The Central Plateau soils are very porous, usually exceeding 70% of TP for Ferralsols [7].

### 4.3. Chemical Attributes

Regarding assortment complex analysis (Table 2 in the Section 3), Group 1 soils (P1, P2, and P3) had low values for chemical attributes, confirming the high degree of weathering [2–4]. Group 2, P4 and P5 had similar values for chemical attributes. However, alluvial sediments from other rocks could have formed the base of these soils due to deposition and removal sequences. Ferralsols showed low cation-exchange capacity (CEC) and base saturation (V) below 37%. High saturation by Al$^{3+}$ (m%) characterizes them as dystric (associated with V < 50%) and aluminic soils [25,28]. All results demonstrated these soils' low natural fertility, except for agricultural use [7].

Nevertheless, P5 achieved a V value of only 8.5 due to higher aluminum and hydrogen (H$^+$+Al$^{3+}$) contents and the highest CEC. Thus, CEC, V, and SB values were considered average for local soils [6,7,33]. These authors verified Ferralsols from FAL with high saturation by aluminum (m values exceeded 20%), expressing a high degree of evolution. As a result, almost all the exchangeable bases were leached, acidifying the soil and making it unsuitable for most crops [30].

Group 2 soils have the same source materials but are younger soils. The percentage of Al$^{3+}$ saturation was higher than in the other soils, either because of the acidic pH or because of the complexation of SOM [3]. In P4, the V value was about 41%, making it the most fertile soil from the toposequence studied. P5 is highlighted by significant variations of decomposition of SOM, which represents a histic horizon (values above 80 g.kg$^{-1}$), according to [25]. P5 stood out for different values for saturation by aluminum (%), cation exchange capacity CEC (maximum 13.44 cmolc dm$^{-3}$), and phosphorus (2.72 cmolc dm$^{-3}$) provided by SOM and clay particles.

### 4.4. Mineralogical Composition

Chemical and mineralogical composition is an indicator of pedogenesis and serves for soil taxonomy, according to the IUSS Working Group WRB, 2015 [25]. Therefore, clay

minerals allow us to assess the weathering intensity in soils [2]. The X-ray diffraction profile can help efficiently model the pedological composition [4,30].

Figure 3 shows the results of the XRD mineralogy analyses, which show that Group 1 soils are essentially composed of kaolinite (Kt), gibbsite (Gb), hematite (Ht), goethite (Gt), and, predominantly, quartz (Qz). Nonetheless, in P3 (Figure 3), small peaks of illite (Il) minerals, with a 2:1 clay–mineral ratio, were detected in the clay fraction, suggesting a slightly lower evolution degree than P1 and P2. We observed more prominent features of Gb in P1 and P2. Novais et al., 2021, 2023, [6,7] found similar results in this study region.

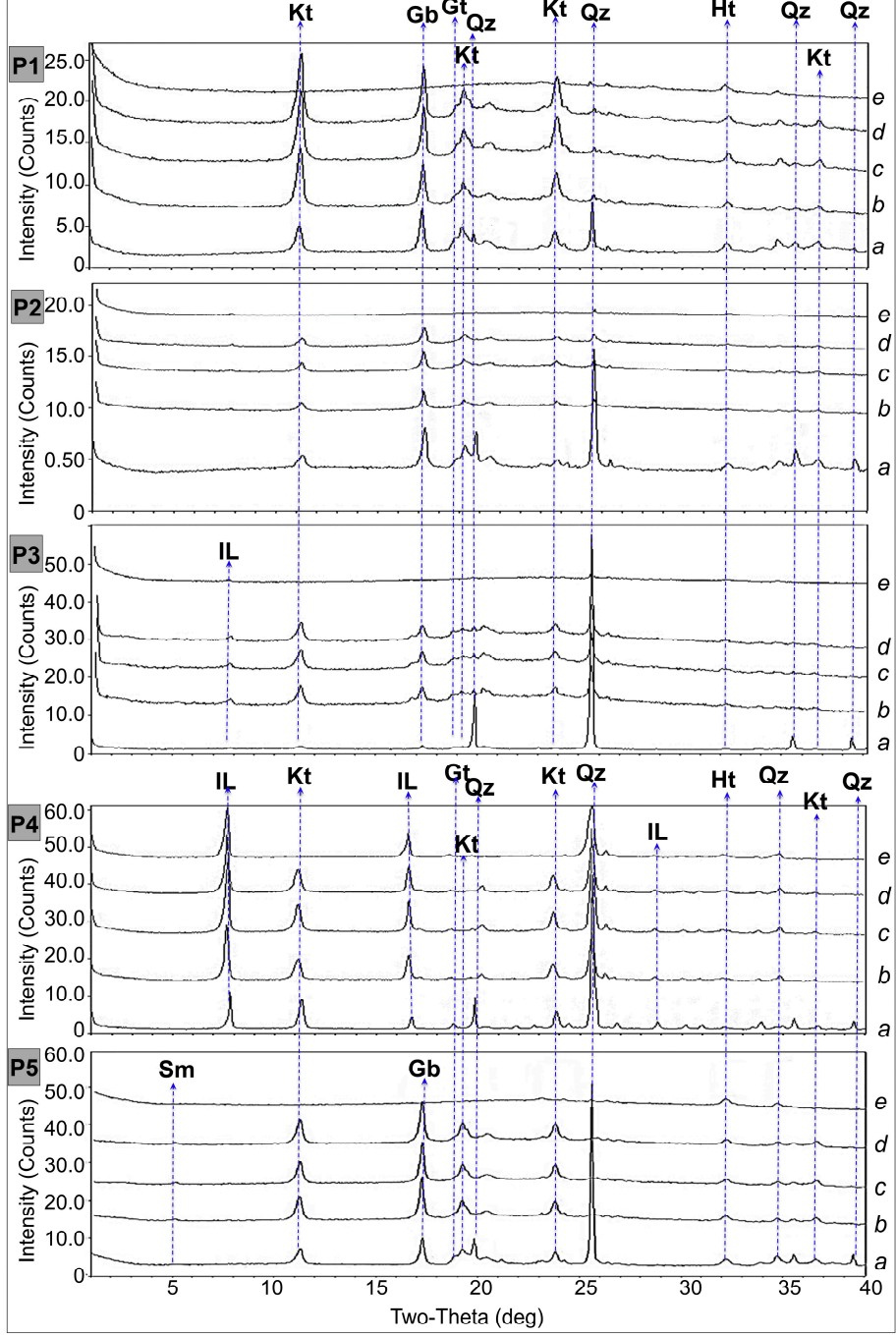

**Figure 3.** X-ray diffractograms of diagnostic horizons from the toposequence studied. Treatments: (a) total; (b) clay; (c) ethylene glycol; (d) glycerin cold; (e) heating. The arrows show the alignments of mineral features through the graphs: (Kt) kaolinite; (Gb) gibbsite; (Ht) hematite; (Gt) goethite; (Qz) quartz; (IL) illite; and (Sm) smectite.

The indicative Gb values were slightly lower in P3 due to the concretionary horizon, at a depth of about 1.25 m, which makes Ferralsol behave in a slightly less evolved fashion in soil than Ferralsol P1 and P2. Terra et al., 2018 [3] stated that aluminum hydroxides (Gb) and Fe oxides (Gt and Ht) in the soil solution indicate a high degree of weathering under conditions of total desilication.

### 4.5. Morphological Interpretation of Reflectance Spectrum

Through the spectral signatures of the soils studied, we have identified variations in the intensity of reflection and absorption of compounds at their specific intervals, showing heterogeneity in degrees of weathering [2–4]. Morphological interpretation of the reflectance spectrum helped us differentiate aspects of soils' spectral curves. According to Liu et al., 2020 [12], VNIR and SWIR reflectance spectroscopy can be a suitable method to derive various pedomorphogeological information, such as mineralogy, texture, SOM content, and other attributes.

Pearlshtien and Ben-Dor, 2020 [15] observed that the first spectral absorption features from Ht are broader than those of Gt in the VNIR range. These spectral features are associated with the reddish or yellowish colors of Ferralsols, respectively [6]. It is worth noting that the lower reflectance in Ferralsols is also due to the clayey texture and the influence of SOM and Fe oxides, which are considered opaque minerals and are obliterated by SOM, with absorption features from 0.6 to 1.0 μm [7].

### 4.5.1. Morphological Interpretation of Spectra from Group 1 Soils

As expected, Fe oxide features were more prominent in Group 1 soils, in which the ferratilization process is intense. In this sense, spectral signatures of horizons A and B in P1 (Figure 4a) show absorption features typical of Fe oxide presence, clay-mineral ratios of 1:1 (Kt), and aluminum hydroxide (Gb). Obliteration by SOM had a lower effect on the A horizon in P1 (30.35 g.kg$^{-1}$) compared to the varying A horizon in P2 (40.5 g.kg$^{-1}$) (Figure 4b) and the A horizon in P3 (53.38 g. kg$^{-1}$), which showed a higher level of obliteration than P2 and P3 (Figure 4c).

A highlighted feature of Fe oxides, Kt and Gb can be observed in P1, where Ht predominates over Gt and Gb is greater than Kt, representing an increase in the soil evolution degree. As an outcome, this demonstrates an advanced ferratilization process because of the highest spectral reflectance factor on the B horizon (0.57), showing an ascending shape in VNIR and a flat one in SWIR that descends after 2.0 μm, with a high albedo caused by a fine texture and less SOM, which is typical of tropical soils. Higher albedo and mineral features occurred in all subsurface horizons due to greater SOM in their respective topsoil [13].

The Kt (1.4 and 2.2 μm) and Gb (2.265 μm) features can be observed in the spectral curve of the B horizon spectra in P2 (Figure 3, previous), where the Kt/Gb ratio is lower than in P1, which has a slightly higher degree of evolution. As exhibited in P1, the Ferralsol spectral curve pattern displays an ascending shape between the 0.35 and 1.3 μm wavelengths, changing to a flattened aspect from 1.3 to 1.85 μm, then decreasing from 1.85 to 2.5 μm. Novais et al., 2023 [7] reassert that Haplic Ferralsols typically exhibit spectral curves similar to those of Rhodic Ferralsols. However, they have a lower reflectance factor.

We found features of Gt at the 0.48 μm wavelength, as well as from 0.90 to 0.95 μm, which explain the yellowish (7.5YR) colors when this oxide predominates, as observed in P3 (Figure 4a). It also showed a strong absorption feature in the spectral range from 0.35 to 1.0 μm, possibly caused by high levels of SOM, which lowers the reflectance and obscures the typical features of Kt, Gt, Gb, and Fe oxide features [13]. The presence of Kt has an absorption feature in the 1.4 and 2.2 μm bands, characterized by a slight inflection on the left side related to water adsorbed by the soil, which was more prominent in P3. Furthermore, the shape of Ferralsol in P3 diverged from the other Ferralsols in P1 and P2 due to the higher SOM content and the concretionary horizon, decreasing the reflectance and smoothing features, as stated by [11].

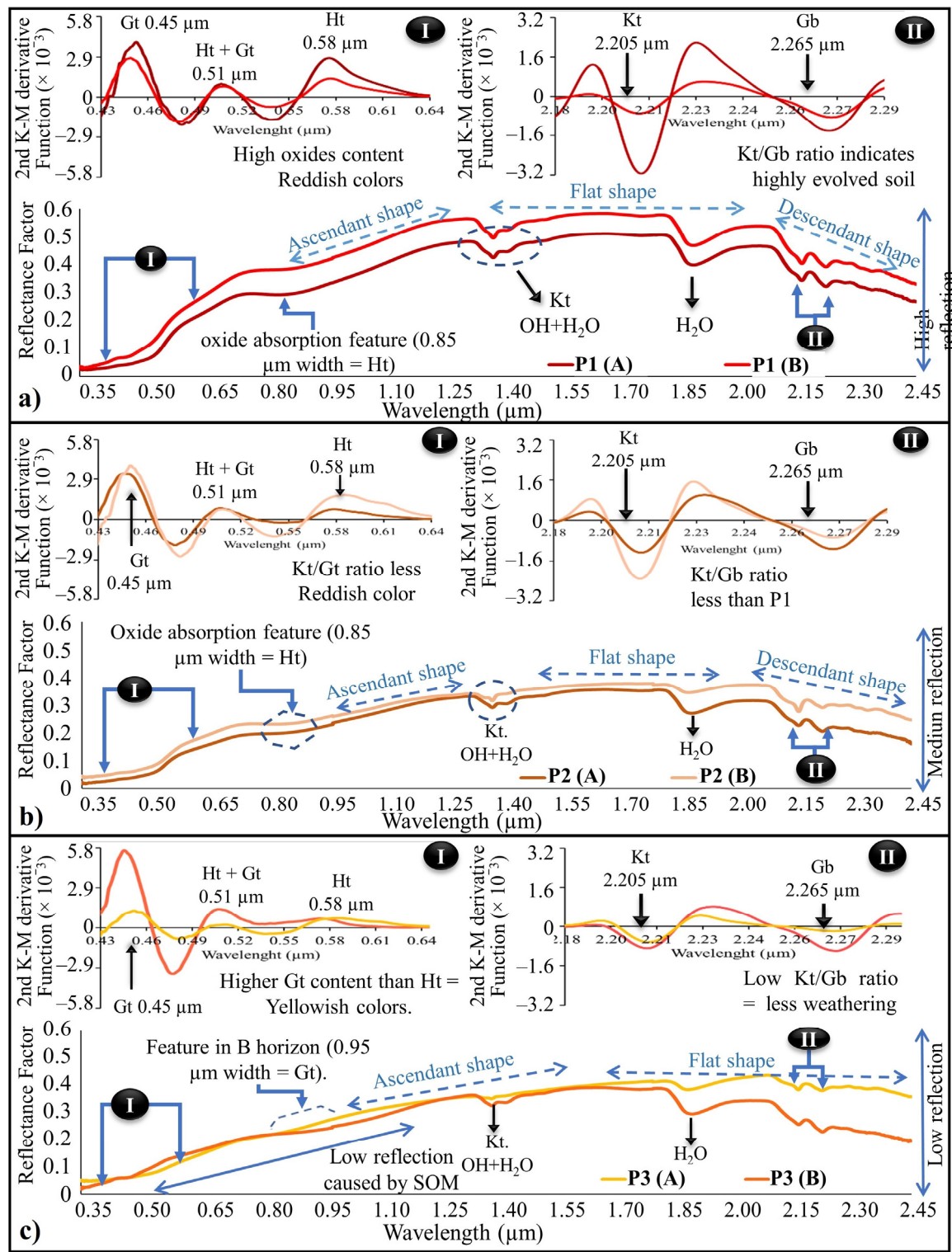

**Figure 4.** Group 1—Morphological Interpretation of the Soil Spectrum: (**a**) Spectral signatures of P1 horizons highlighting their main features, (**b**) Spectral signature of P2 horizons highlighting their main features, and (**c**) Spectral signatures of P3 horizons highlighting their main features. Ellipses in black (I and II) signify enlarged areas analyzed with the Kubelka-Munk second derivative function; (P) soil profile; (A) surface horizon; (B) subsurface horizon; (Kt) kaolinite, (Gb) gibbsite, (Ht) hematite, (Gt) goethite, and (SOM) soil organic matter.

### 4.5.2. Morphological Interpretation of Spectra from Group 2 Soils

In both P4 and P5 spectra (Figure 5a,b, respectively), the curve exhibited a significant 'V-shaped' absorption at 1.4 μm with a shoulder on the left side, typical of Kt. On the right side of this wavelength, an absorption feature was highlighted more than in other soils. This feature is attributed to the 2:1 mineral ratio, including minerals such as illite, observed in the P4 diffractogram (Figure 3, previously shown in the Section 4.4).

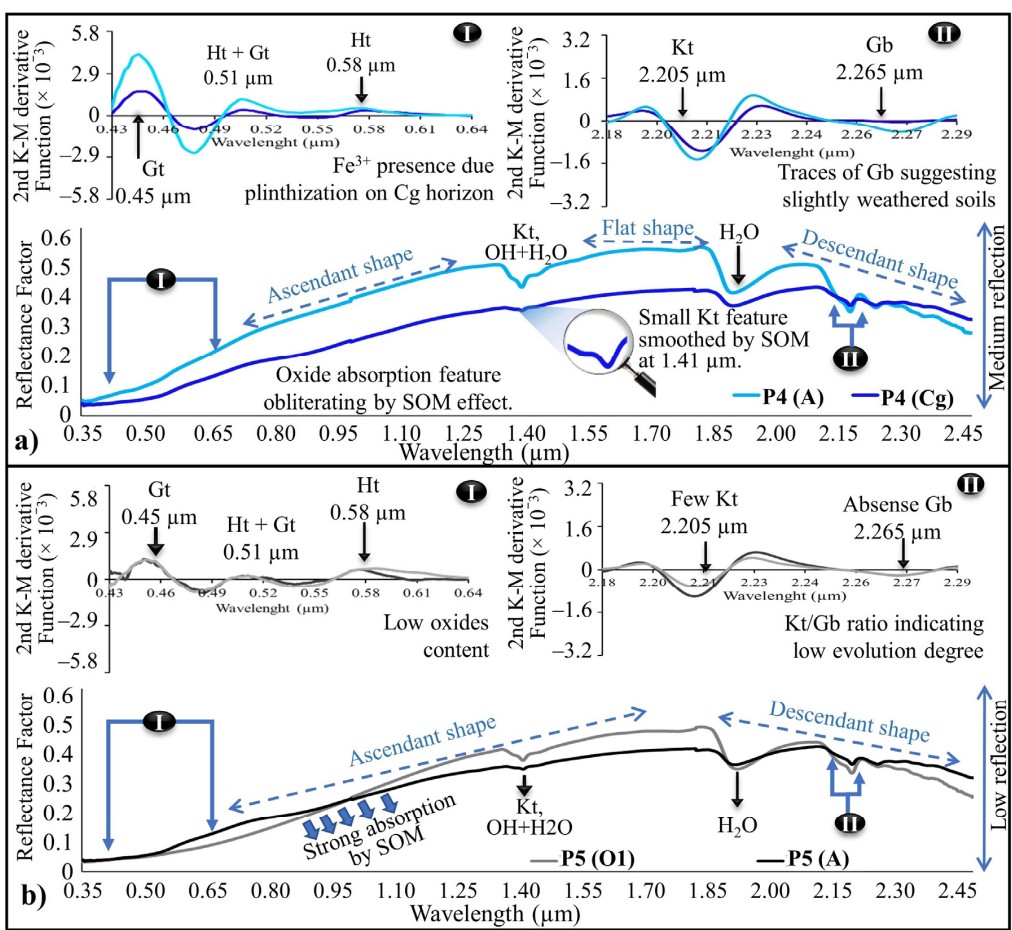

**Figure 5.** Group 2—Morphological Interpretation of the Soil Spectrum. (**a**) Spectral signatures of P4 horizons highlighting their main features, and (**b**) Spectral signatures of P5 horizons highlighting their main features. Ellipses in black (I and II) signify enlarged areas analyzed with the Kubelka-Munk second derivative function; (P) soil profile; (A) surface A horizon of Gleysol; (Cg) subsurface horizon of Gleysol; (O1) surface organic horizon of Histosol; (Kt) kaolinite, (Gb) gibbsite, (Ht) hematite, (Gt) goethite, and (SOM) soil organic matter.

Demattê et al., 2014 [13] found that the absence of oxidized Fe traces in the spectral curve characterizes the hydromorphism process, considering that iron is predominantly present in its reduced form. Moreover, spectral signatures of P4 showed weak imprints of Gt (0.48 and 0.9–0.95 μm) and Ht (0.58 and 0.85 μm) left by Fe oxides from the plinthization process. Such oxides are being obliterated by SOM content in soil solution.

Terra et al., 2018 [3] characterized Gleysols in plain relief with variations in particle size. The range in spectral behaviors observed in Group 2 soils (P4 and P5) is caused mainly by SOM content, soil particles, and mineralogical composition. P4 showed an inflection at 2.105 μm related to Kt absorption and a slight presence of Gb at 2.3 μm. The plinthization pedogenic process occurred at the $C_{glei}$ horizon, explaining the Fe oxide features. The 2:1 clay mineral presence was detected at 1.45 μm from a small step on the right side of the water absorption feature.

The last profile, P5 (Figure 5b), demonstrated curves with very low reflectance factor values in the VNIR range influenced by SOM, causing an upward flattening in the interval between 0.35 at 1.4 μm [3]. Beyond that, we observed a 2:1 clay mineral feature through slight inflections promoted by Kt (2.105 μm) presence and hydroxyl groups represented by Gb (1.4, 1.9 and 2.265 μm). The Gb feature (2265 μm) was almost null, without traces of Fe oxide features in the A horizon from 0.35 to 1.2 μm. P5 had the lowest albedo values in the VNIR range (0.35 and 1.35 μm) and the maximum values (0.33) in SWIR at the A horizon. Blackish colors observed with the naked eye were verified in this spectral range [6]. We correlate the soil spectra to the main soil attributes in the next section.

### 4.6. Correlation between Soil Attributes and Wavelengths

The mean soil spectral curve was submitted to Pearson's correlation with soil attributes positioned along the wavelengths from 0.35 to 2.5 μm under the 2nd derivative function of Kubelka–Munk to enhance the spectral features (Figure 6). This technique can assist researchers in making decisions about the best spectral bands for implementing predictive models and improving the existing low-performance models [6,7,10]. Liu et al., 2020 [12] also estimated pedological properties using predictive models by correlating soil attributes and spectra.

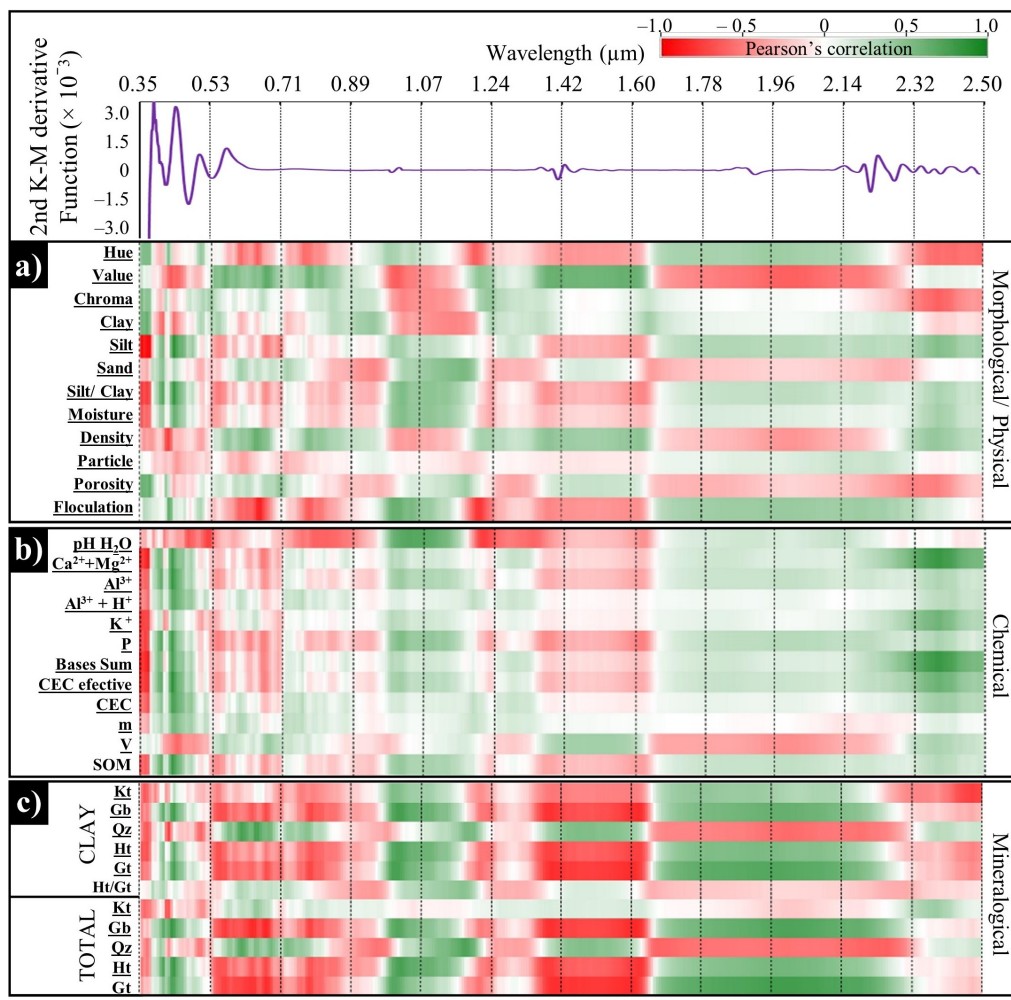

**Figure 6.** Pearson's correlation between Vis-NIR-SWIR wavelengths under Kubelka–Munk function and (**a**) morphological and physical soil attributes, (**b**) chemical soil attributes, and (**c**) mineralogical composition in clay and total treatments: (CEC) cation exchangeable capacity; (V) base saturation; (SOM) soil organic matter; (Kt) kaolinite; (Gb) gibbsite; (Ht) hematite; (Gt) goethite; (Qz) quartz; (IL) illite; and (Sm) smectite.

Regarding the color attributes (Figure 6a), we observed that the value parameter has an opposite correlation to hue, presenting a high correlation in spectral ranges in SWIR. Chroma showed an almost null correlation, especially from 1.60 to 2.32 μm, varying to the negative relationship at the end of SWIR (from 2.32 to 2.50 μm). Poppiel et al., 2019 [10] modeled tropical soil attributes and considered VNIR regions to be better for mapping soil color due to the contrast present in these bands of the spectrum and the Munsell Color Chart, 1992 [26]. Soil hue obtained a positive correlation between 1.60 and 2.32 μm but was negative for VIS intervals and SWIR (2.32 to 2.50 μm). Correlation for value had a behavior opposite to hue, which reinforces the validity of this method since they are inversely proportional on the Munsell Color Chart [26]. Chroma had almost zero correlation from 1.60 to 2.32 μm, varying to a negative relation at the end of SWIR (from 2.32 to 2.50 μm). The higher reflectance oscillation in VNIR than SWIR is caused by electromagnetic interaction with the soil compounds [13].

Figure 6a also exhibits physical attributes. Texture has a demonstrated relationship within SWIR, varying more along soil spectra. Clay was correlated positively at 0.53 (green) and 0.89 μm (NIR) and negatively in SWIR (1.07 and 2.32 μm). Nevertheless, silt correlated negatively in VNIR and positively at 1.07 μm and SWIR (1.70 μm), with a high positive correlation at the end of SWIR (2.32 to 2.50 μm). The sand fraction was positively correlated at 0.71 μm (red) and in SWIR (1.07 μm) and negatively between NIR and SWIR. While the S/C ratio and Ms had similar behavior, Ds showed a positive correlation at 0.89 and 1.24 μm. Our study demonstrated that the total soil porosity from Group 2 varied from 50.98% at the $C_{glei}$ horizon of P4 to 84.03% at the O horizon of P5. The SWIR region showed itself to be more representative of TP because it exhibited a high negative correlation. The flocculation degree varied between Ferralsols (78.15 to 91.89 g g$^{-1}$) and was correlated positively in SWIR and negatively in VNIR. A similar correlation was found to clay content but was inversely proportional to sand content. The S/C ratio presented a higher relationship, showing that the 0.35 to 0.53 μm and 2.32 to 2.50 μm spectral ranges were better for studying these soil attributes. Jaconi et al., 2019 [9] succeeded in predicting soil texture in their studies using a multispectral resolution orbital sensor. Poppiel et al., 2019 [10] successfully predicted soil texture classes through different wavelengths of reflectance spectroscopy at various soil depths. Both studies indicated the SWIR as the best for spectral modeling.

Considering the correlations of spectra with chemical and mineralogical attributes (Figure 6b,c), there was a highly positive correlation at the end of SWIR to exchangeable bases ($Ca^{2+}$, $Mg^{2+}$, and $K^+$), SB, CEC at pH 7, t, m, and V. As mentioned in Section 4.2, SOM content entails soil darkening in the VIS interval [6]. The positive correlation between SWIR absorption bands and SOM content can explain the absence or presence of features of the minerals Gb and Kt at 2:1 ratios due to the obliteration caused by this attribute on soil spectra. Pearlshtien and Ben-Dor, 2020 [11] stated that visible and near-infrared spectroscopy, combined with other properties, can predict soil's physical and chemical attributes. In this study, the higher correlation found between pH and $H_2O$ shows a different spectral hub that can be used in the attribute prediction model. Soil acidity occurs in tropical soils; low $Ca^{2+}$ and $Mg^{2+}$ and high $Al^{3+}$ contents characterize it [4]. This condition comes from parental material, as we relate in the next topic.

Figure 6c shows the correlation between the soil spectral signature wavelengths and mineralogy features, represented by intensity peaks in XRD. Such a relationship was expressed by the Total and Clay treatments, and there were more prominent peaks and higher variations of Kt content present in the clay fraction of the studied soils, increasing the correlation to positive or negative in the Clay treatment. We also noted a low correlation between spectra and the total Kt treatment. However, at the beginning of VIS (0.35 to 0.53 μm), there was a positive correlation with Kt under the Clay treatment, demonstrating a strong correlation positive in VIS and SWIR. Qz features in the Clay and Total treatments had an inverse correlation with spectra between 0.53, 0.71, and 1.07 μm. Schoeneberger, 2012 [27] stated that Qz is the most abundant mineral on Earth and directly influences

soil texture. The oxides (Ht, Gt, and Gb) showed a positive relation, mainly from 0.53 to 0.89 μm and 1.42 to 1.60 μm, with further negative effects from 1.60 to 2.32 μm for both treatments. These minerals are directly related to the soil evolution degree because they demonstrate the last stage of desilication, when most of the exchangeable bases are leached, leaving the sesquioxides from Fe and Al that are responsible for the reddening of soils, which explains the correlation of these along spectra, especially in VNIR [13]. Therefore, the amount of all these characteristics affects the spectral behavior of soils, making them identifiable by spectral patterns.

### 4.7. Pedogenic Process Assessment by Spectral Response of Soil

The subdivision of the soil groups by the weathering stage was essential to observe differences in the spectral data of the soils. We followed them from changes in attributes such as the mineralogy of clay fraction, the Ht/Gt ratio, and the toposequence. Similarly, chemical attributes could contribute to pedogenic assessment, but SOM content negatively influenced it, impairing specific feature identification.

As stated by Novais et al., 2021, 2023 [6,7] and Poppiel et al., 2019 [10], finer texture and higher Fe oxide and SOM contents decrease reflectance. Such attributes were noted on the profile studied and could relate to the pedogenic process. The S/C ratio showed values far below 0.6, which is the limit for Ferralsols of the region.

Soil colors from Group 1 ranged from dark red to red-yellowish and were attributed to a higher concentration of Fe oxides supplied from source material formed by pelitic metasediment rocks rich in ferruginous cement [6,7,10]. In turn, features caused by the presence of Ht are highlighted between the 0.58 μm and 0.85 μm wavelengths.

Firstly, the ferratilization process occurred for all profiles from Group 1, with slightly lower intensity in P3 due to the concretionary layer acting as a barrier to water infiltration [7]. Spectral curves of P1, P2, and P3, consisting of Ht, Gt, Kt, and Gb, represent mineralogy resulting from a high stage of desilication and concentration of sesquioxides (Fe oxides and Al hydroxides).

The hydromorphism process, considered an addition process, formed P4 differently from Ferralsols, which are formed by the remotion processes characterizing evolved soils. The gleization process in the $C_{glei}$ horizon was observed in the spectral curve of this soil, characterized by traces of Fe oxides and a high concentration of SOM in the A horizon, which are the main characteristics of hydromorphic soils under a reducer environment [3]. The conditions of P4 indicated a degradation process impeding its ability to provide ecosystem services and consequently diminishing healthy soil characteristics.

The presence of 2:1 clay–mineral ratios shows the lowest degree of evolution compared to Ferralsols. Traces of Fe oxides were related to the plinthitization process observed in the field in-depth Cg horizon. Horizon A in P4 and P5 was marked by the influence of SOM when we analyzed the VNIR region of the P4 and P5 spectra, which obliterated the presence of oxides and other mineral characteristics.

### 4.8. Individualization Capacity of Soil from the Spectral and Physical Data

We performed the HCA (Figure 7) using the spectral data and texture of the soils as an attempt to validate the clustering of the soils studied according to their weathering level, as described by Terra et al., 2018 [3]. Two simple dendrograms, showing soil grouping (Figure 7) from five categories on the vertical and horizontal axes, plotted the mineralogy values. In this case, we utilized the Ht/Gt obtained by XRD and K–M reflectance factors of soils for clustering analyses.

P1 and P2 (Rhodic Ferralsols) had more similarities concerning other profiles. In the same way, P4 and P5 (hydromorphic soils) were joined. P3 (petroplinthic, Haplic Ferralsol) was closer to the P4 and P5 groups. The formation of three groups is highlighted right in the middle of the dendrogram. According to our data table, it was observed that the P3 is more outstanding among Ferralsols. In this sense, Poppiel et al., 2019 [10] stated that specific spectral features of soils can discriminate them.

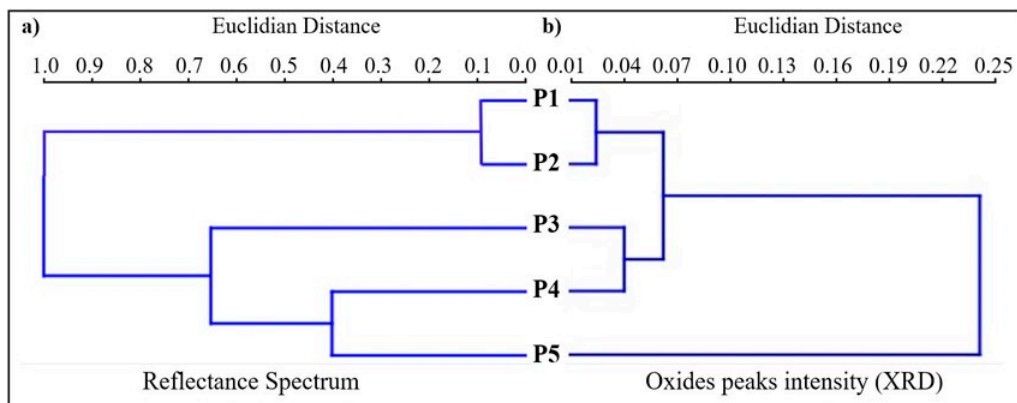

**Figure 7.** Dendrograms of hierarchical cluster analysis based on (**a**) K–M function reflectance spectra and (**b**) soil oxides (clay content).

However, the grouping related to oxide peaks in spectral curves showed a significant separation of P5 (Haplic Histosol) due to SOM content, which achieved higher levels compared to other soil classes. Concerning the individual soil classes, we observed other profiles forming groups from the toposequence base level to P1–P2, followed by the P3–P4 cluster. According to Terra et al., 2018 [3], this behavior suggests that the pedoenvironment in P3 exerted influence on its attributes in a way near to P4 (Haplic Gleysol).

Considering the pedogenic processes occurring in the five soil profiles, we could observe the conformity of analyses with the field truth: the ferratilization occurred in the first group, which were highly weathered Ferralsols kept on the top of the plateau. The weathering decreases and the slope increases until the appearance of Haplic Ferralsols with a concretionary horizon, which hinder water infiltration, promoting changes in the pedoclimate. The second group presented a Haplic Gleysol formed by the gleization process in a reductiomorphic environment. The humification pedogenic process generated a Haplic Histosol at the toposequence bottom.

These results revealed the potential of remote sensing data in pedological assessments from the perspectives of efficiently supporting the monitoring and analysis of spatial distributions of soil classes and attributes on a large scale [2–4]. Several researchers cited throughout this study have already discussed the possible applications in agricultural engineering or environmental studies. All agree that integrated knowledge (holistic) can drive the future of the soil sciences. Using sensors at the orbital level can enhance the ability to revisit soil monitoring, guiding actions to conserve or exploit this non-renewable natural resource [6,7,10].

## 5. Conclusions

We observed the spectral bands of VNIR-SWIR correlating with soil properties, assessing soil spectral behavior according to its distinctive properties. Thus, we demonstrated that traditional analyses and geotechnologies can assist in assessing pedomorphogeological relationships. In addition, we showed the influence of pedogenic processes on spectral responses in tropical soils from the Central Plateau region of Brazil. These results can assist pedologists in understanding soil origins, properties, management, usage, and other covariate interactions.

The morphological interpretation of the reflectance spectrum identified changes in the pedoclimate of hydromorphic soils, including the occurrence of a concretion layer constituted by petroplinthites, which shows the paleopedogenic evolution of this soil. On the one hand, this observation allows us to verify that the higher oxide content and finer texture are more determinant for highly evolved soil studies, demonstrated by the ferratilization process influencing the soil spectral signatures. On the other hand, lower iron oxide content and high values of organic matter had a higher impact on hydromorphic soils, which are less-evolved soils.

Moreover, hierarchical cluster analysis of soil reflectance spectroscopy demonstrated the ability to discriminate between soils. Other non-parametric statistical approaches with a robust database should be used to improve qualitative and quantitative investigations about pedogenic processes in all soil classes worldwide. However, periodic measurements are required, making orbital sensing data an alternative because it can serve as a continuous data source for soil monitoring.

**Author Contributions:** Conceptualization, J.J.N., R.R.P., M.P.C.L. and J.A.M.D.; data curation, J.J.N., R.R.P. and J.A.M.D.; formal analysis, J.J.N.; funding acquisition, J.J.N., R.R.P., M.P.C.L. and J.A.M.D.; investigation, J.J.N. and R.R.P.; methodology, J.J.N., R.R.P. and J.A.M.D.; project administration, J.J.N., M.P.C.L. and R.R.P.; resources, J.J.N., M.P.C.L., J.A.M.D. and R.R.P.; software, R.R.P., J.J.N. and J.A.M.D.; supervision, M.P.C.L. and J.A.M.D.; validation, J.J.N. and M.P.C.L.; visualization, J.J.N. and R.R.P.; writing—original draft, J.J.N.; writing—review and editing, J.J.N. and R.R.P. All authors have read and agreed to the published version of the manuscript.

**Funding:** This work was funded by CAPES—Coordenação de Aperfeiçoamento de Pessoal do Ensino Superior (Coordination for Improvement of Higher Education Personal); University of Brasília—UnB; The Federal District Research Support Foundation (Fundação de Apoio à Pesquisa do Distrito Federal—FAP/DF): official notice 16733.78.29498.26042017 and the São Paulo State Research Support Foundation (Fundação de Apoio à Pesquisa do Estado de São Paulo—FAPESP): process number 2021/05129-8.

**Data Availability Statement:** The datasets related to physical, chemical, morphological, and spectral soil analyses, as well as the shapefiles archives generated in this study, are available on request directly from the first and second authors or the Geoprocessing and Pedomorphology Laboratory—Geoped (geoped.unb@gmail.com) or Geotechnologies in Soil Science—GeoCiS (geocis@usp.br).

**Acknowledgments:** The first author thanks the co-author for assisting in fieldwork and laboratory analyses and methodological and theoretical support. On behalf of the Geotechnologies in Soil Science group (GeoCiS, https://esalqgeocis.wixsite.com/english, accessed on 30 May 2023) from ESALQ/University of São Paulo, we acknowledge the supervisors, particularly the fourth author, for spectral readings and methodology instruction. It is worth thanking the Faculty of Agronomy and Veterinary Medicine from the University of Brasília–UnB for support and providing structure. In advance, we thank the editors and anonymous revisors who contribute to this text improvement.

**Conflicts of Interest:** The authors declare no conflict of interest.

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
