# Peer review of "VNIR-SWIR Spectroscopy, XRD and Traditional Analyses for Pedomorphogeological Assessment in a Tropical Toposequence"

_agriengineering, doi:10.3390/agriengineering5030098_

Round 1
Reviewer 1 Report
This paper aims to assess the relationships between pedomorphology and geology using VNIR-SWIR reflectance spectroscopy and X-Ray diffractometry analyses. The study presents detailed information on morphological and physical attributes, chemical attributes, mineralogical composition, and more. The experimental results are impressive, and the discussion is comprehensive. These findings can greatly assist pedologists in understanding soil origin, properties, management, usage, and interactions with other factors. However, there are still some issues that need to be addressed. The specific comments are as follows:
1) Please provide an explanation for the unequal depth of excavations shown in Figure 2. Additionally, please clarify the unclear sign in Figure 3 and consider redrawing it.
2) Please modify the subtitle 4.1.2. Morphological and Physical Attributes (line 182, Page 6) to 4.1.1. Morphological and Physical Attributes. The analysis section of 4.1.2. Chemical attributes is comparatively weak compared to 4.1.1. Morphological and Physical Attributes.
3) Remote sensing monitoring is a rapidly growing technique that enables large-scale monitoring and analysis of spatial distributions. Please expand on the potential applications and possibilities in this field and discuss how it can be integrated with future research.
This paper aims to assess the relationships between pedomorphology and geology using VNIR-SWIR reflectance spectroscopy and X-Ray diffractometry analyses. The study presents detailed information on morphological and physical attributes, chemical attributes, mineralogical composition, and more. The experimental results are impressive, and the discussion is comprehensive. These findings can greatly assist pedologists in understanding soil origin, properties, management, usage, and interactions with other factors. However, there are still some issues that need to be addressed. The specific comments are as follows:
1) Please provide an explanation for the unequal depth of excavations shown in Figure 2. Additionally, please clarify the unclear sign in Figure 3 and consider redrawing it.
2) Please modify the subtitle 4.1.2. Morphological and Physical Attributes (line 182, Page 6) to 4.1.1. Morphological and Physical Attributes. The analysis section of 4.1.2. Chemical attributes is comparatively weak compared to 4.1.1. Morphological and Physical Attributes.
3) Remote sensing monitoring is a rapidly growing technique that enables large-scale monitoring and analysis of spatial distributions. Please expand on the potential applications and possibilities in this field and discuss how it can be integrated with future research.
Author Response
Dear Reviewer,
We consider that your contributions and questions have improved the intelligibility of our article. It was extensively revised following your and other referees’ appointments. All alterations were marked in red throughout the text.
The answers to your questions about specific comments follow attached.

Reviewer 2 Report
This manuscript utilizes several methods to evaluate the pedomorphogeology of a tropical toposequence in Brazilian Midwest. Through analyzing a large number of experimental results, it was determined that all soil classes are correlated with their corresponding reflectance spectra and primary soil attributes. It introduces the feasibility of using VNIR-SWIR spectroscopy, X-ray diffraction, and traditional analysis methods for assessing the pe-domorphogeological relationships. The approach applied in the article have practical application value. However, there are some problems that need to be further considered. The specific opinions are as follows:
1. The part of the content mentioned in the abstract is specific experiment results and processes, which should be described in the Introduction of experiment.
2. The abstract mainly describes the feasibility of the methods used in the manuscript. But it does not explicitly state the academic contribution of the work.
3. The wrong format is used in Equation (1), and it looks like an image.
4. The content of Figure 3. is obscured, that is difficult for the readers to recognize.
5. In the manuscript, the authors used a lot of pronouns such as “we” and “our”, which makes the narration less objective.
Minor editing of English language required
Author Response
Dear Reviewer,
We consider that your contributions and questions have improved the intelligibility of our article. It was extensively revised following your and other referees’ appointments. All alterations were marked in red throughout the text.
The answers to your questions about specific comments are as follows:

Reviewer 3 Report
The text discusses how tropical climate conditions influence the evolution of landscapes and the development of highly weathered soils through various pedogenic processes. The study employs a combination of traditional analysis methods and advanced geotechnologies, including VNIR-SWIR reflectance spectroscopy and X-Ray diffractometry (XRD), to investigate the characteristics of soils in a representative toposequence in the Brazilian Midwest.
The researchers conducted field observations and soil sampling, followed by laboratory analyses to determine physical, chemical, spectral, and mineralogical properties of the soils. The classification of soils was done according to the World Reference Basis – WRB/FAO system. By utilizing Pearson's Correlations and Hierarchical Clustering Analysis (HCA), they divided the soil profiles into two groups based on their degree of weathering.
The paper presents findings related to different pedogenic processes in these soil groups. For instance, highly weathered Ferralsols were found on the plateau's top, showcasing the process of ferratilization. The soil weathering decreased as the slope increased, leading to the emergence of Haplic Ferralsols with a concretionary horizon that affected water infiltration and induced changes in pedoclimate. The second group contained a Haplic Gleysol formed by gleization in a reductiomorphic environment, and a Haplic Histosol generated through humification was observed at the bottom of the toposequence.
The study revealed strong correlations between the spectral oxide features and X-ray diffraction peaks, which were used to validate the previous soil grouping using Hierarchical Clustering Analysis (HCA) based on oxide content and mineral composition. The research demonstrates the significance of VNIR-SWIR spectroscopy, XRD, and traditional analyses in understanding pedogenic processes and their influence on soil properties. The authors also note the need for periodic measurements and suggest the potential of orbital sensing data to provide continuous soil monitoring information.
The researchers utilize the reflectance spectrum to interpret changes in the pedoclimate (soil environment) of hydromorphic soils, particularly highlighting the presence of a concretion layer made up of petroplinthites. This layer serves as evidence of the historical evolution of the soil. The study reveals that higher levels of iron oxide content and finer soil texture play a crucial role in the study of highly evolved soils, as exemplified by the influence of the ferratilization process on spectral signatures. Conversely, soils with lower iron oxide content and higher organic matter content have a greater impact on hydromorphic soils, which are less evolved.
Additionally, the passage mentions the successful application of hierarchical cluster analysis to discriminate between different types of soils based on their reflectance spectroscopy. It also suggests that employing non-parametric statistical methods with a comprehensive database could further enhance qualitative and quantitative investigations into pedogenic processes across various soil classes worldwide.
However, the study acknowledges that regular measurements are necessary for accurate soil monitoring. In this context, the passage proposes that orbital sensing data can offer an alternative by providing a continuous source of information for monitoring soil conditions over time.
In conclusion, I agree with the publication of this work.
Author Response
Dear Reviewer,
We consider that your contributions and questions have improved the intelligibility of our article. It was extensively revised following your and other referees’ appointments. All alterations were marked in red throughout the attached text.
Round 2
Reviewer 2 Report
No more comments
Minor editing of English language required